# Quorum Signaling Molecules: Interactions Between Plants and Associated Pathogens

**DOI:** 10.3390/ijms26115235

**Published:** 2025-05-29

**Authors:** Xi Zheng, Junjie Liu, Xin Wang

**Affiliations:** State Key Laboratory for Conservation and Utilization of Bio-Resources in Yunnan, Yunnan University, Kunming 650091, China; zhengxi138@163.com (X.Z.); jakeliu_ok@163.com (J.L.)

**Keywords:** quorum sensing, AHL, farnesol, ascarosides, plant defense, inhibitors, QQ

## Abstract

The morphogenesis and defense evolution of plants are intricately linked to soil microbial community dynamics, where beneficial and pathogenic bacteria regulate ecosystem stability through chemical signaling. A microbial communication mechanism known as quorum sensing (QS), which affects population density, virulence, and biofilm formation, substantially impacts plant development and immune responses. However, plants have developed strategies to detect and manipulate QS signals, enabling bidirectional interactions that influence both plant physiology and the balance of the microbiome. In this review, QS signals from bacteria, fungi, and nematodes are systematically examined, emphasizing their recognition by plant receptors, downstream signaling pathways, and the activation of defense responses. Most significantly, attention is given to the role of fungal and nematode QS molecules in modulating plant microbe interactions. By elucidating these communication networks, we highlight their potential applications in sustainable agriculture, offering novel insights into crop health management and ecosystem resilience.

## 1. Introduction

Plants coexist with a variety of soil microbiomes, some of which are pathogens that reduce plant fitness, while others are beneficial bacteria that promote plant growth and stress tolerance [1]. These microbial communities employ sophisticated communication systems known as QS, which allow them to coordinate group behaviors based on population density through the production and detection of quorum signaling molecules (QSMs) [2,3]. Crucially, plants have evolved the capacity to decipher or even sabotage these QS signals, turning microbial communications into a battlefield for survival. While bacterial QS mechanisms (e.g., N-acyl homoserine lactones, AHLs) are well-characterized, recent studies reveal that fungal farnesol derivatives and nematode ascarosides similarly orchestrate host–pathogen interactions, albeit through less understood pathways [4,5]. Though less well-characterized than their bacterial counterparts, they play equally pivotal roles in pathogen–host plant communications and virulence strategies. Table 1 details QS molecules associated with certain plant-associated microorganisms. These signals are no longer viewed merely as microbial tools but as inter-kingdom mediators that directly manipulate plant immunity and development. For instance, plants decode long-chain AHLs (e.g., oxo-C14-HSL) as danger cues to prime systemic resistance, while hijacking short-chain AHLs to enhance root growth; this is a double-edged sword that microbes exploit in colonization [6,7]. 

As signaling molecules accumulate at threshold concentrations, they bind to specific receptors and initiate coordinated gene expression programs that regulate the behaviors of microbial communities, including bioluminescence, biofilm formation, virulence production, antibiotic production, and the establishment of symbiotic relationships [8,9,10,11]. Furthermore, microbial QSMs promote plant growth and root architecture and activate plant immune defenses. It has been proposed that some AHL molecules, including oxo-C14-HSL and oxo-C8-HSL, may be recognized by *Arabidopsis thaliana* as danger signals that activate immune responses and enhance resistance to bacterial pathogens such as *Pseudomonas syringae* [6,7]. Through this primed state, cells can respond better to pathogens, reinforced by lignin and callose deposition in the cell walls [12].

The plant surveillance system is enhanced by sophisticated countermeasures such as the production of QS mimic molecules and enzymes, which can disrupt microbial communication and decrease virulence [13]. These fundamental insights into QS systems have led to the development of quorum sensing inhibitors (QSIs), which offer promising new avenues for the prevention and control of plant diseases [14]. Unlike conventional antimicrobials, QSIs interfere with microbial communication rather than killing pathogens, thus limiting opportunities for resistance to develop [8]. An example would be garlic-derived QSI compounds that inhibit *Pseudomonas aeruginosa* virulence and block signal reception by *Xanthomonas* pathogens via synthetic analogs of AHL [15]. This strategy provides protection against plant pathogen infections while reducing bacterial resistance, offering an alternative method of controlling plant diseases.

Increasing understanding of QS mechanisms across kingdoms (bacteria, fungi, and nematodes) indicates their potential application in sustainable disease management practices. Three key avenues deserve attention: (1) pharmacological interference through QSI compounds (e.g., garlic-derived inhibitors); (2) ecological modulation through QSM mimics synthesized in plants; (3) microbiome engineering based on interspecies QS crosstalk [16,17]. In this review, we provide an overview of the QSMs for bacteria, fungi, and parasitic nematodes associated with plants. It summarizes the effects of QS on plants and their feedback interactions while exploring QS applications related to plant diseases. By undertaking these studies, new avenues are expected to emerge for the development of safer chemical alternatives, providing theoretical support for sustainable agriculture. Future research should focus on elucidating the molecular mechanisms of QS systems and developing more efficient and specific QSIs to address the challenges associated with plant diseases.

## 2. QS Definition and Core Mechanisms

QS is a cell-to-cell communication mechanism used by microorganisms to coordinate group behaviors by producing and detecting specific signaling molecules known as QSMs [18]. There are several categories of QSMs, as shown in Figure 1 and Table 1. In bacteria, QSMs consist of autoinducers such as AI-1 (N-acyl homoserine lactones), AI-2 (furanosyl borate diester), AI-3, and DSF (diffusible signaling factor) [19,20]. Less common types, such as oligopeptides and furanones, are crucial for pathogenesis [21]. Among fungi, farnesol derivatives are identified as key QSMs and, among plant-parasitic nematodes, ascarosides serve as pheromone-like QSMs. Despite being less well characterized than their bacterial counterparts, nematode and fungal QSMs play equally important roles in pathogen–host plant communication and virulence, as shown in Figure 1 and Table 1.

When the microbial population density reaches a certain threshold, QSMs accumulate and bind to receptors (intracellular or membrane-bound receptors on other cells), triggering the expression of related genes [8]. This process is responsible for controlling various microbial community behaviors, including the secretion of virulence factors, biofilm formation, antibiotic production, and the establishment of symbiotic relationships [9,10]. QSMs can be directly recognized by host cells, while directly impacting host physiological functions [8]. It is worth noting that microbial QSMs play a significant role not only in plant growth and root architecture, but also in the infection process. This signaling occurs through the regulation of QSM-dependent effector release during infection. For instance, bacterial QS signals, such as oxo-C14-HSL or oxo-C8-HSL, may be cleverly utilized by the roots of *Medicago truncatula* and *A. thaliana* seedlings as danger signals to initiate the expression of immune genes. This regulation involves transcription factors, G-protein, and calcium signaling, as well as the salicylic acid/oxylipin pathway, thereby enhancing resistance to *P. syringae* pv. *Tomato* (*Pst*) [6,7].

## 3. Major Classes of QSMs and Their Functions

### 3.1. Bacterial QSMs

AHLs are the dominant QS signal molecules in Proteobacteria, which are widely used by pathogens including *Ralstonia solanacearum*, *P. syringae*, and *Xanthomonas oryzae* and mutualist *Rhizobium* species [22,23,24,25]. In terms of structure, the conserved HSL ring facilitates signal recognition by binding to LuxR-family receptors, while the variable acyl chain (C4-C18 with hydroxyl/keto groups) defines the specificity of functions [26,27,28]. In this way, AHLs mediate flexibility in QS regulation across different bacterial species and hosts. For example, plant cells are passively diffused with short-chain C4-HSL to stimulate root growth, and long-chain 3-oxo-C12-HSL can activate immune responses by binding to receptors [29].

The Autoinducer-2 molecule is a QS molecule produced by Gram-positive and Gram-negative bacteria, catalyzed by the LuxS enzyme. As its core structural component, 4,5-dihydroxy-2,3-pentanedione (DPD) undergoes spontaneous rearrangement to give rise to a variety of DPD derivatives known as the AI-2 pool [30]. A number of derivatives of DPD are available, including the boron-containing derivative S-THMF-borate and the non-borated derivative R-THMF [31]. Plant pathogenic bacteria (e.g., Pectobacterium) use S-THMF-borate to coordinate their infection. However, R-THMF is commonly associated with symbiotic or mutualistic behaviors, such as nitrogen-fixing nodulation by rhizobia (e.g., *Rhizobium*). Occasionally, certain bacteria, such as *Salmonella*, can detect both derivatives simultaneously [32,33,34].

The DSF molecules (cis-11-methyl-dodecenoic acid) contain cis-2-unsaturated fatty acids with specific carbon chain lengths and double-bond configurations, which are primarily found in Gram-negative bacteria such as *Xanthomonas campestris*. *campestris* (*Xcc*), *Burkholderia cenocepacia* (*Bcc*), and *P. aeruginosa* [35]. Due to these signaling molecules, bacterial behavior is finely modulated, affecting their symbiotic and pathogenic interactions with plants. Through the secretion of several virulence factors, such as exopolysaccharide (EPS), extracellular cell wall-hydrolyzing enzymes, and glucan by the DSF, *Xanthomonas* enhances its pathogenicity in rice [36]. The DSF quorum sensing signal molecule in wild rapeseed *X. campestris* regulates a variety of biological functions, including three types of metabolic adjustments, to adapt to high-population-density environments, inhibition of biofilm formation, and enhanced expression of pathogenicity-related genes in *X. campestris*.

Gram-positive bacteria utilize short peptides (AIPs) (5–17 residues), either cyclic or linear, which contain a conserved cysteine (positions 3–5 from the C-terminus) that forms a thioester linkage with the terminal residue. These AIPs, well-documented in species such as *Staphylococcus*, *Streptococcus*, and *Bacillus*, function through two-component systems (e.g., histidine kinase-response regulators). Notably, the plant growth-promoting rhizobacterium (PGPR) *Bacillus subtilis* utilizes the ComX AIP to activate the comQXP pathway and regulate root colonization [36]. Aside from the QSMs mentioned above, the pyrazinone-derived autoinducer AI-3, which is usually found in pathogens such as *E. coli* and *Vibrio cholerae*, remains controversial in plant-associated bacteria [37].

### 3.2. Fungal QSMs

QS phenomena in fungi are less well-known than in bacteria, but certain QS phenomena and regulatory molecules have been clearly identified in some fungi. For example, farnesol is an active QS molecule that regulates various processes such as morphogenesis, biofilm development, mating, drug efflux, and apoptosis in *Candida albicans* [38]. In yeast, aromatic alcohols such as phenylethanol (PheOH) and tryptophol (TrpoH) drive filamentation [38,39]. Additionally, in *Fusarium oxysporum*, pheromones are involved in spore germination in response to cell density [40]. Unlike the structurally conserved AHL in bacteria, fungal QSMs are primarily alcohols, lipids, and other small molecules which are not structurally conserved. Fungi coordinate pathogenicity-related behaviors at the individual to population level through QS regulation, including germination, colony morphogenesis, sporulation, and biofilm formation [41,42].

### 3.3. Parasitic Nematode QSMs

Nematodes use pheromones to detect complex environmental situations and thereby modulate their behavior, population density and development, a mechanism analogous to bacterial QS [43,44,45]. The ascaroside signaling molecules consisting of an ascarylose core linked to a fatty acid side chain undergo modifications that result in highly conserved yet functionally diverse molecules [46]. Multiple nematode species, including parasitic ones, contain these molecules and can modulate host gene expression and immune responses [47], as shown in Figure 1 and Table 1. To date, over 300 ascaroside variants have been identified in more than 20 nematode species [48]. In parasitic nematodes, ascaroside#18 (Ascr#18) is particularly prevalent, characterized by an ascarylose core coupled to a C11 fatty acid. The application of Ascr#18 to *Arabidopsis* results in resistance to a variety of plant pathogens and insects [49,50].

**Table 1 ijms-26-05235-t001:** Diverse effects of quorum signaling molecules (QSMs) on plants.

QSMs	Producing Pathogens	Plant	Functions/Effect	Reference
C4-HSL(RhlI), 3-oxo-C12-HSL (LasI)	*P. aeruginosa*	*Arabidopsis*	growth promotion	[51]
3-hydroxy-C4-HSL	*Vibrio harveyi*	Tobacco	plant resistance	[52]
C6-HSL	*P. aeruginosa*	*A. thaliana*, wheat	root growth, enhances cereal crop resistance to pathogens and abiotic stress	[29,53]
C10-HSL to C14-HSL	*P. aeruginosa*	Barley and *Arabidopsis*	resistance toward biotrophic and hemibiotrophic pathogens	[54,55]
C_8_-HSL, C_7_-HSL	*Castellaniella defragrans*, *Cryobacterium* sp.	*Mortierella alpine* A-178	colonization	[56]
C6-and C8- HSL	*S. liquefaciens*, *Pseudomonas putida*	Tomato	ISR-like response	[57]
3-OH-C10-HSL	*Acidovorax radicis N35*	Barley	colonization of roots	
3-oxo-C14-HSL	*Sinorhizobium meliloti*	*M. truncatula*	nodulation in roots	[58]
C12-HSL and C16-HSL	*Agrobacterium vitis*	*M. truncatula,* *A. thaliana*, *Hordeum vulgare*	AHL-priming	[59]
oxo-C12-HSL, oxo-C16-HSL	*Ensifer meliloti* *P. aeruginosa*	*M. truncatula*	auxin-responsive and flavonoid synthesis; mimicking QS secretion	[60]
oxo-C14-HSL	*S. meliloti*, *Ensifer melilot*	*Arabidopsis* barley, wheat, and tomato *M. truncatula*	AHL-priming for agriculture root nodulation in *M. truncatula*	[61,62]
1-aminocyclopropane-1-carboxylate indole-3-acetic acid	*Burkholderia phytofirmans*	*Phaseolus vulgaris*	endophytically colonizes and promotes plant growth, forms symbiotic nodules and fix nitrogen	[63]
furanosyl borate diester (AI-2)	*Pasteurella, Photorhabdus*, *Haemophilus,* and *Bacillus*	Zoosporic plants	promoting plant infection	[64]
pyrazinone derivative (AI-3)	*E. coli, Shigella* sp. and *Salmonella* sp.	Animals	virulence	[65]
cyclic dipeptides	*H. marmoreus*	*Arabidopsis*	triggers plant immunity	[66,67]
CAI-1	*Vibrio cholerae*	/	/	[68,69]
(*R*)-3-OH PAME, (*R*)-3-OH MAME	*R. solanacearum*	Tomato, tobacco, and potato	pathogenicity	[70,71]
indole-3-acetic acid	*Azospirillum*, Rhizobacteria	*Citrus cinensis* *Arabidopsis*	root formation	[72]
*N*-3-oxo-hexanoyl-homoserine	*M. truncatula*	*Arabidopsis* and wheat	enhances salt tolerance, primary root elongation	[73]
3-oxo-C6-HSL	*Pantoea stewartii*	Mung beans, *Arabidopsis*	plant pathogen	[74]
3-oxo-C14-HSL	*S. meliloti*	Mung beans barley, wheat, and tomato	nitrogen-fixing symbiont, plant immunity	[75]
3-oxo-9-cis-C16-HSL	*S. Meliloti*, *P. aeruginosa*	Mung beans	nitrogen-fixing symbiont; induces auxin response and flavonoid synthesis	[76,77]
3-hydroxy-7-cis-C14-HSL	*Rhizobium leguminosarum*	Mung beans	nitrogen-fixing symbiont	[78]
9-cis-C16-HSL	*Sinorhizobium melioti*	*Medicago*	nitrogen-fixing symbiont	[79,80]
farnesol	*Candida albicans,* *Trichoderma harzianum*	Tomato	plant defense, regulates morphogenesis, biofilm development, sporulation, mating, drug efflux, and apoptosis,	[38,41,81]
Phenylethanol, tryptophol	Yeast	*A. thaliana* and tomato	drives filamentation	[38,39]
2-ethyl-1-hexanol	*F. oxysporum*	*A. thaliana* and tomato	enhances plant growth	[40]
α-factor	*Saccharomyces cerevisiae,* *Aspergillus fumigatus*	Tomato	infection	[42]
ascr#1, ascr#3, ascr#9, ascr#10, ascr#18, oscr#9	*M. Incognita*, *M. javanica*, *M. hapla*, *H. glycines*, *Pratylenchus brachyurus*	*Arabidopsis,* tomato, potato and barley	resistance to plant pathogens	[49,50]

Note: The majority of AHL-producing isolates from the plant rhizospheres belong to the genera *Pseudomonas*, *Rhizobium*, *Serratia*, *Burkholderia*, *Erwinia*, and *Pantoea*.

## 4. Perception of and Responses to QSMs

Plants can simultaneously perceive QSMs, activate immune responses, or adjust their physiological state to combat pathogen invasions [62]. Studies in *Arabidopsis* have provided a great deal of insight into the mechanisms of plant responses to QSMs, as shown in Figure 2.

### 4.1. Molecular Mechanisms of QSM Perception in Plants

Plants employ distinct recognition strategies for different classes of QSMs based on their physicochemical properties and molecular structures. For bacterial AHLs, chain length governs perception mechanisms. The short-chain AHLs (C4/C6-HSL) passively diffuse across plasma membranes due to their small size and hydrophobicity, subsequently interacting with cytosolic histidine kinases like HK1 [82]. Medium-chain AHLs (C8/C10-HSL) are detected through membrane-localized receptors, including GPCR (G-protein-coupled receptor) complexes (GCR1-GPA1) and RLK (receptor-like kinases) (Cand2/Cand7), which bind these molecules via extracellular domains with precise acyl chain length selectivity [83]. Long-chain AHLs (C12/C14-HSL) require active transport by ABC transporters (e.g., ABCG40) and subsequent recognition by lectin receptor kinases (LecRK-I.9) through PAN-Apple domains [84,85,86]. Non-AHL bacterial signals like XcDSF hijack host sterol biosynthesis to alter membrane properties, disrupting receptor clustering (e.g., FLS2). Fungal QSMs are decoded through specialized systems: farnesol binds oxysterol-binding protein (ORP) homologs, while ergosterol/squalene engage GPCR-like receptors with steroid-binding domains. Nematode ascarosides (e.g., ascr#18) are specifically captured by the NILR1 receptor ectodomain, followed by peroxisomal β-oxidation to generate shorter derivatives [49]. These interactions highlight a sophisticated “molecular fingerprinting” system where plants discriminate QSM structural features among lactone rings (AHLs), isoprenoid tails (terpenoids), and cyclohexenone cores (ascarosides), further initiating context-dependent responses.

### 4.2. Plant Responses to AHLs

AHLs can trigger distinct physiological responses in plants depending on their carbon chain lengths. Short-chain AHLs (e.g., C4-HSL, C6-HSL) promote plant growth and root development by modulating auxin (IAA), cytokinin (CK) signaling, and the cell cycle [87]. In roots, short-chain AHLs upregulate auxin biosynthesis genes (e.g., YUCCA) and transporters (e.g., PIN), stimulating primary root elongation, lateral root formation, and root hair development to enhance nutrient uptake [88]. They also accelerate meristematic cell proliferation by regulating cell division-related genes (e.g., CYCD) [6,89]. Although short-chain AHLs show potential as microbial fertilizer enhancers, these molecules can mildly suppress immune-related gene expression (e.g., PR1), thereby reducing defense-associated metabolic costs and diverting resources toward growth [90].

AHLs with long chains (e.g., C12-HSL, C14-HSL) promote disease resistance by priming plant immunity through epigenetic modifications and preactivation of defense-related genes [86,91], triggering early defense signals, including MAPK cascade activation (e.g., MPK3/MPK6), calcium fluctuations, and ROS (reactive oxygen species) and NO (nitric oxide) bursts, establishing an immune-alert state [92]. Epigenetically, long-chain AHLs induce histone modifications (e.g., H3K4me3, H3K9ac) and DNA methylation changes, loosening the chromatin structure of defense-related genes (e.g., PR1, WRKY53) to establish an immune memory. As a result, antimicrobial metabolites (phenols, phytoalexins) are accumulated [93]. In this manner, the SA (salicylic acid) and JA (jasmonic acid) pathways are prioritized, while growth-related pathways (e.g., gibberellin, GA) are moderately suppressed [94].

Medium-chain AHLs (e.g., C8-HSL, C10-HSL) interact with plants in a concentration- and receptor-dependent manner, facilitating growth while regulating immunity. The Arabidopsis GPCR receptors are involved in the regulation of root growth and biomass accumulation at low concentrations of 1 μM 3OC6-HSL or 10μM 3OC8-HSL [84,95], likely by coordinating auxin (IAA), cytokinin (CK) pathways, and strigolactone signaling via the transcriptional factor AtMYB 44. By contrast, high concentrations (>10µM) trigger immune-related gene expression (e.g., PR1, WRKY53), bolstering pathogen resistance [96]. Due to this dual nature, medium-chain AHLs are useful in regulating plant–microbe ecological interactions, although their precise mechanisms of regulation must be further explored.

Unlike AHL signals, AIPs (autoinducing peptides) mediate intracellular communication but do not freely diffuse across cell membranes. Their release into the extracellular space is facilitated by specialized oligopeptide transporters, primarily ABC transporters [97,98]. AIPs primarily function in animal intestinal barriers, while their role in plants remains scarce and has been reported mainly in *Bacillus*. Oligopeptides do not diffuse freely through the cell membrane; therefore, the cells need two-component phosphorelay cascades, consisting of a membrane-bound receptor/sensor histidine kinase protein with an intracellular response regulator, to sense extracellular oligopeptides [99].

During infection, plants perceive these bacterial cues: *X. campestris* (Xcc)-XcDSF (100–1000 μM) elicited immune responses (callose deposition, PR-1 expression, HR-like cell death) in *Nicotiana*, *Arabidopsis*, and *rice*, enhancing resistance to *X. oryzae* [100]. At lower concentrations (25 μM), mimicking early infection, XcDSF suppressed PTI (pattern-triggered immunity) by hijacking host sterol biosynthesis. This alters plasma membrane (PM) properties, disrupting FLS2 receptor clustering and endocytosis, desensitizing plants to PAMPs such as flagellin [101]. Additionally, XcDSF remodels the cell wall–PM–actin cytoskeleton (CW–PM–AC) continuum, impairing immune coordination [101]. It also enhances cellulose biosynthesis, mechanically perturbing formin-mediated actin dynamics during PTI [35]. This spatiotemporal regulation highlights complex bacteria–plant crosstalk, where *Xanthomonas* exploits host structures for virulence.

### 4.3. Plant Responses to Fungal QSMs

Exogenous farnesol inhibits the growth of *tomato* and bean plants while upregulating SA-related defense genes, acting as a toxic MAMP/PAMP [102]. Similar immune responses are observed in tomatoes exposed to Trichoderma mycotoxins, including harzianum A [103]. Unlike farnesol, terpene analogs (squalene, ergosterol) induce only JA/ET-related genes and not SA-related genes [103]. Trichoderma-derived farnesol elevates SA signaling, while ergosterol/squalene dominance shifts responses toward JA/ET. A combination of these metabolites serves as a fungal MAMP or PAMP, and their relative concentrations determine how plants respond to them. A high level of farnesol promotes SA pathways, while a high level of ergosterol/squalene favors JA/ET pathways [104]. This highlights the nuanced role of fungal terpenes in shaping host immunity [103].

Similarly, fungal oxylipins are better understood and known to regulate developmental processes including cell growth, sexual and asexual spore differentiation, apoptosis, and pathogenicity [104,105]. The various oxylipins present in plants function as molecular signals that regulate growth and development, senescence, sex determination of reproductive organs, defense against biotic and abiotic stresses, and programmed cell death [106]. It is becoming increasingly clear that oxylipins, such as jasmonates, operate primarily by influencing signal crosstalk with other hormones. In general, host-derived oxylipins (e.g., jasmonates in plants) facilitate resistance to attack by fungal pathogens [107].

### 4.4. Plant Responses to Nematode QSMs

Plants recognize nematode-derived ascarosides (e.g., Ascr#18) through specific receptors such as StNILR1. This recognition triggers immune responses in plants, analogous to their responses to bacterial flagellin (Flag22), lipopolysaccharides (LPS), or fungal chitin and β-glucans. The plant metabolizes ascarosides (e.g., Ascr#18) into a mixture of short-side-chain derivatives, including Ascr#8, Ascr#10, Ascr#1, and Ascr#9. The metabolized short-chain ascarosides are then secreted by plant roots, activating immune responses that subsequently inhibit nematode infection. Additionally, plants mediate the signaling of both brassinosteroids (BR) and ascarosides (e.g., Ascr#18) through interactions with coreceptors such as StBAK1. This synergistic mechanism ensures a balanced regulation between immune defense and growth responses in plants [108,109].

## 5. Strategies Used by Plants to Disrupt Pathogen QSMs

Pathogens use the QS system to coordinate group behaviors, such as secreting virulence factors or launching collective attacks against host plants. However, over time, plants have evolved a suite of defensive mechanisms to disrupt bacterial QS systems [110], primarily involving three levels of inhibition strategies: degradation of QS compounds, interference with receptors, and suppression of QS synthesis [17].

### 5.1. Metabolic Modification of Pathogen QSMs

The chemical structure of AHLs is inherently unstable, making them susceptible to hydrolysis or acylation in the environment or by plant-released hydrolytic enzymes such as acylases or lactonases [111,112,113]. Moreover, plant-derived exudates (alkaloids, sugars, hormones, polysaccharides, proteins, and lactones) exhibit notable anti-plant pathogenic bacteria activity [114]. *A. thaliana*’s growth effects depend on AHL amidolysis by a plant-derived fatty acid amide hydrolase (FAAH), yielding l-homoserine, which can encourage plant growth at low concentrations by stimulating transpiration [111]. Moreover, plants can edit parasitic nematode-derived Ascr#18 pheromones to produce other ascaroside groups involved in developing defense mechanisms [108].

### 5.2. QS Mimics Enabling Receptor Interference

QSIs that target AI signaling molecules are primarily AHL-lactonases (for example, autoinducer inactivation A, AiiA hydrolyzes 3-oxo-C14-HSL in rice blight resistance), oxidoreductases, neutralizing antibodies (such as mAb AP4-24H11 binding to *S. aureus* AIP-1) and small molecules such as vanillin (which blocks tomato C8-HSL receptor binding) [115,116,117]. Inhibiting the QS system is accomplished by inactivating signaling molecule synthases, neutralizing AIPs with antibodies, modifying or degrading the signaling molecules, and so on [65,118,119].

Quorum sensing inhibitors (QSIs) targeting autoinducer signaling molecules primarily include (i) AHL-lactonases (for example, AiiA hydrolyzes 3-oxo-C14-HSL and attenuates the virulence of Erwinia carotovora); (ii) oxidoreductases, which either oxidize the acyl chain of AHLs or reduce 3-oxo-AHLs to their corresponding 3-hydroxy-AHL counterparts; (iii) neutralizing antibodies, such as mAb AP4-24H11 binding to S. aureus AIP-1; and (iv) small molecules such as vanillin (which blocks tomato C8-HSL receptor binding). Savirin inhibits AgrB to prevent AIP production in *S. aureus* and provides an alternative approach for halting signal molecule production at its source. As a result of these different inhibitors, QS is disrupted in a variety of ways, including signal degradation, receptor binding interference, direct signal neutralization, and synthase inactivation.

Plants produce a variety of plant QS mimics that are structurally similar to bacterial QS signals and interfere with the target QS system [120], including phenolic compounds such as coumaric acid and caffeic acid. Mimicking compounds can bind competitively to LuxR-type receptor proteins or induce conformational changes in the receptors, blocking recognition of natural AHL signal molecules [121,122]. The halogenated furanones derived from the red alga *Delisea pulchra* are among the earliest and best-characterized AHL mimics. They promote the degradation of the AHL-LuxR complex and inhibit QS-regulated behaviors in *Serratia liquefaciens* [123]. The mimics can also act as agonists that trigger premature expression of QS genes, potentially reducing bacterial virulence by disrupting QS-driven functions [124]. Although rosmarinic acid does not have the structural characteristics of AHLs, it induces premature quorum sensing responses both in vitro and in vivo, indicating its potential as an agricultural product [125]. *Flavanone naringenin* also demonstrates plant-derived QS interference by suppressing virulence factors (e.g., pyocyanin and proteases) in *P. aeruginosa* PAO1 [126]. Additionally, seed exudates from *M. truncatula* contain approximately 20 compounds capable of interfering with LuxR-type QS biosensors, demonstrating the diversity of plant strategies for rhizosphere microbiome regulation [123]. Beyond their pathogen-suppressing ability, these molecules may also enhance plant growth and influence root architecture, highlighting their dual ecological significance. Some plant metabolites, such as clove oil [127], exhibit concentration-dependent regulatory effects. At low concentrations, they can partially activate the QS pathway but disrupt gene expression (known as “pseudo-activation”), while at high concentrations they completely suppress QS pathway activity [128].

Moreover, plants can systematically disturb bacterial cell-to-cell communication by secreting QS signal-degrading enzymes or synthesizing potent inhibitors, ultimately impairing coordinated virulence factor expression and biofilm formation. Specific examples demonstrate this sophisticated interference: *Medicago sativa* (alfalfa) secretes l-canavanine, an arginine analog that inhibits EPS production in *S. meliloti* [129], while *Combretum albiflorum* releases the flavonoid flavan-3-ol catechin, which suppresses QS-regulated virulence factors in *P. aeruginosa* PAO1 [130].

### 5.3. Quorum Quenching

Quorum quenching (QQ) is a crucial defense strategy in plant–microbe interactions, where plants recruit symbiotic bacteria with QQ functions (such as *B. subtilis* and *Microbacterium testaceum*) to interfere with the QS system of pathogenic bacteria [131]. In these symbiotic bacteria, various QS signal-degrading enzymes are secreted, including AiiA from *Bacillus* spp., AttM from *Agrobacterium tumefaciens*, and AiiM from *M. testaceum* [132,133], which inhibit pathogen virulence by degrading QSMs such as AHL. It has significant practical value in agriculture, particularly in managing and controlling diseases such as *Erwinia carotovora’s* soft rot and *Pectobacterium carotovorum*’s potato rot [134]. Moreover, QQ bacteria can enhance systemic resistance (ISR) in plants by activating jasmonic acid (JA) and ethylene (ET) signaling pathways, thereby forming a broader defense network [135]. To date, several typical QQ proteins have been found in biocontrol bacteria, such as *Bacillus thuringiensis*, which degrade AHL molecules produced by pathogenic bacteria. Researchers have discovered that the enzyme-producing *Lysobacter enzymogenes* OH11 induces the type IV secretion system (T4SS) to quench *Pseudomonas fluorescens*’ AHL-based communication system [136].

## 6. QS Plant Immunity and Sustainable Solutions

### 6.1. Metabolic Responses

Plant immune priming induced by AHLs, referred to as AHL-priming, enhances and accelerates defense responses without directly blocking pathogen invasion [137]. By reprogramming the plant’s epigenetic system, the plant becomes “alert”, which induces stronger resistance against subsequent pathogen attacks [138]. This denotes a physiological state in which plants can activate their defense responses in a faster and stronger manner in response to a triggering stimulus.

AHLs induce metabolic reprogramming through the activation of MAPK cascades (mitogen-activated protein kinase signaling pathways), Ca^2+^ signaling, and WRKY/MYB transcription factor networks. As a result, multilayered metabolic pathway rearrangements occur [139], which produce a “defense metabolome” containing glucosinolates, antimicrobial fatty acids, and so on [140]. Additionally, plant parasitic nematodes, such as root-knot nematodes and cyst nematodes, may also influence plant immune signaling by secreting NAMPs (nematode-associated molecular patterns). Evidence suggests that certain ascarosides activate the plant MAPK signaling pathway, thereby triggering PTI immunity [141,142].

### 6.2. Ecological Impacts

Physiological and ecological aspects of AHL-mediated metabolic regulation are complex [143]. For example, *Arabidopsis* activates MAPK signaling through degrading C6-HSL, while tobacco overexpressing lactonase AiiA exhibits enhanced resistance to soft rot disease [144]. A symbiotic relationship between legumes and nitrogen fixation genes allows specific molecules such as C4-HSL to activate rhizobial nitrogen fixation genes while suppressing the QS of pathogenic bacteria [91]. Further studies have revealed that flavonoids from medicinal plants such as *Scutellaria baicalensis* inhibit QS by chelating AHLs. This study provides insight into plant–microbe coevolution and suggests that plants can influence the rhizosphere microbiota through the regulation of metabolites [145]. Bacterial QSMs may significantly influence plant systemic resistance by modulating SAR (salicylic acid pathway) and ISR (jasmonate/ethylene pathway). By inducing JA/ET, pathogenic AHLs can be recognized as molecular patterns that activate SAR, whereas beneficial bacterial QSMs enhance ISR through JA/ET stimulation [146].

### 6.3. Agricultural Applications

An in-depth understanding of plant-mediated AHL regulation will enhance plant immunity knowledge and facilitate the development of innovative QS-based precision agriculture, environmental restoration, and anti-infection therapies, leading to breakthroughs in research and application. For example, engineered crops expressing AiiA enzymes show potential in combating green diseases, plant-derived QSIs offer potential as novel anti-infection drugs [147], and plant microbe synergistic metabolism has been shown to degrade AHL pollutants in the environment [148].

It is noteworthy that QS-mediated interactions are not only involved in pathogenic relationships but also in cooperative symbioses between plants and beneficial microbes, such as rhizobia and mycorrhizal fungi. For example, AHLs regulate nodulation factor synthesis in rhizobia, while flavonoids facilitate rhizobial QS activity in plants, resulting in a bidirectional regulatory effect [149]. Plant immunity and QQ play a central role in ecological adaptation; this intricate, cross-species chemical dialogue provides a theoretical basis for novel biocontrol agents or genetically modified crops based on QQ [150]. Future research should focus on deciphering the interaction between QSMs and plant epigenetic modifications (e.g., histone acetylation) as well as engineering high-efficiency QQ microbial strains for precise disease management in agricultural ecosystems through synthetic biology [151].

## 7. Conclusions

QS is an essential signaling mechanism in microbial communities, allowing bacteria and fungi to coordinate behaviors such as virulence, symbiosis, and secondary metabolite production. Plants have developed sophisticated strategies to recognize and respond to microbial QSMs, such as AHLs, through immune priming and epigenetic reprogramming. As a result of this priming, MAPK cascades, Ca^2+^ signaling, and defense-related transcription factors are activated, resulting in the accumulation of antimicrobial metabolites. Nevertheless, plants do not remain passive; they actively interfere with QS either by degrading signaling molecules (e.g., through lactonases) or by producing QSIs (e.g., flavonoids and strigolactones) to disrupt microbial communication. In the fields of agriculture, medicine, and environmental remediation, the exploitation of these interactions has great potential. Bioengineered crops that express QS-degrading enzymes (e.g., AiiA lactonase) could offer provide sustainable disease control, while plant-based QSIs may serve as novel antimicrobial agents. Moreover, controlling beneficial microbial consortia through QS modulation could enhance biocontrol, plant growth promotion, and soil remediation practices. Despite the progress made so far, understanding fungal QS systems, identifying plant QS receptors, and optimizing field applications remain significant challenges. A future research program integrating synthetic biology, omics technologies, and smart farming strategies will be necessary to translate these discoveries into practical, scalable solutions for the promotion of sustainable agriculture and ecosystem resilience.

## Figures and Tables

**Figure 1 ijms-26-05235-f001:**
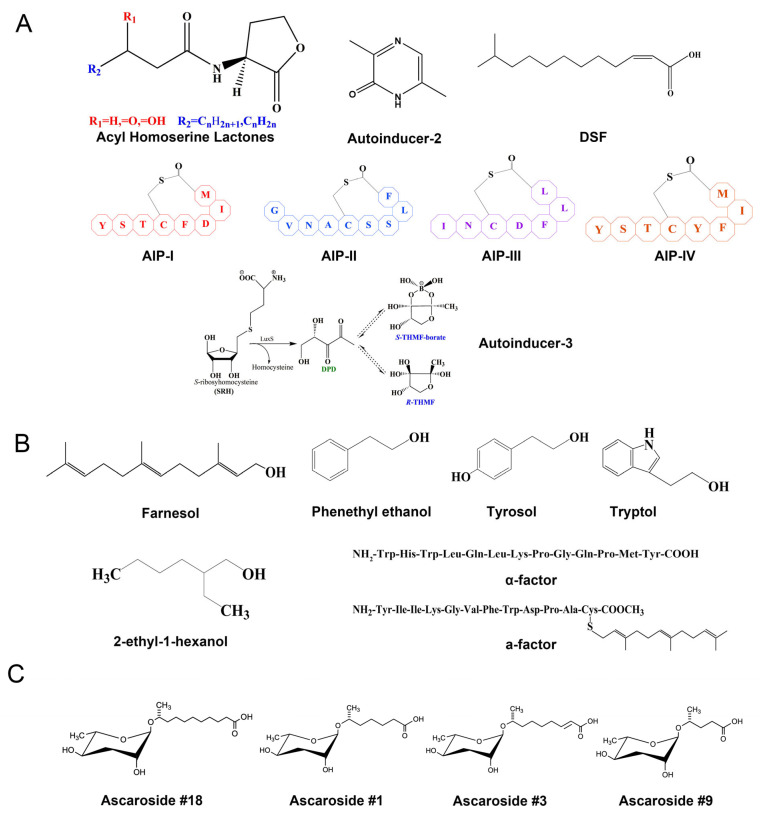
Quorum signaling molecules (QSMs) in bacteria, fungi and parasitic nematodes. (**A**) Representative structures of typical bacterial QSMs; (**B**) Structures of several conserved fungal QSMs; (**C**) Structures of several ascaroside derivatives produced by nematodes. Abbreviations: S-THMF-borate, S-2-methyl-2,3,3,4-tetrahydroxytetrahydrofuran-borate; R-THMF, R-2-methyl-2,3,3,4-tetrahydroxytetrahydrofuran.

**Figure 2 ijms-26-05235-f002:**
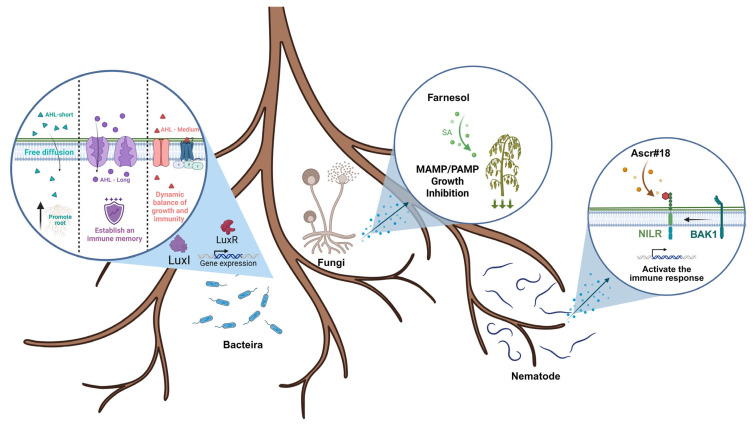
Plant perception of quorum sensing signals: proposed recognition model.

## Data Availability

Not applicable.

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
