# Peer review of "Quorum Signaling Molecules: Interactions Between Plants and Associated Pathogens"

_ijms, 2025, doi:10.3390/ijms26115235_

Round 1

Reviewer 1 Report

Comments and Suggestions for Authors

This is a nice manuscript, but there are also many problems that need to be revised.
1. There are many format errors in this manuscript, including italics, subscripts and references. Please modify them one by one according to the requirements.
2. The emphasis in this paper should not be on the types of quorum sensing signals, so Figures 1, 2 and 3 should be merged. Figures related to the interaction between plants and quorum sensing signals should be added.
3. In Table 1, why only bacteria are counted? quorum sensing signals in fungi and nematodes also should be counted. It is suggested to add this part. In addition, it should be more organized, such as according to the type of QS signals, or according to the type of influence on plants.

Author Response

Reviewer #1: This is a nice manuscript, but there are also many problems that need to be revised.

Response: Dear reviewer 1, thank you very much for your attention to our manuscript and helping our paper processing. It was a great encouragement to receive your positive and valuable comments. We have made a thoroughgoing revision of the manuscript based on your suggestion. Please refer to the revised manuscript for more details.

Once again, thank you very much for your arduous work and instructive suggestions.

Yours sincerely

Xin Wang

Comment 1. There are many format errors in this manuscript, including italics, subscripts and references. Please modify them one by one according to the requirements.

Response: We have carefully double-checked our manuscript and modified the errors you mentioned. Meanwhile, we improve the manuscript language through MDPI Author Services (ID: english-94088) .

Comment 2. The emphasis in this paper should not be on the types of quorum sensing signals, so Figures 1, 2 and 3 should be merged. Figures related to the interaction between plants and quorum sensing signals should be added.

Response: As suggested, we have merged the original Figures 1-3 into a single consolidated figure (now Figure 1), focusing only on essential signal types. Additionally, we added Figure 2 to illustrate the molecular interaction between plants and quorum sensing signals (e.g., receptor binding and immune activation). These changes align the manuscript with its core focus on plant-microbe communication.

Figure 1. Essential quorum sensing signals and their basic microbial mechanisms.

Figure 2. Plant perception of QS signals: Proposed recognition model

Comment 3. In Table 1, why only bacteria are counted? quorum sensing signals in fungi and nematodes also should be counted. It is suggested to add this part. In addition, it should be more organized, such as according to the type of QS signals, or according to the type of influence on plants.

Response: We are grateful for your insightful advice and are pleased to report significant progress in the revised version of our paper, which now includes comprehensive sections on nematodes and fungi. The revisions have been highlighted in red. Please see in the resubmitted manuscript.

Reviewer 2 Report

Comments and Suggestions for Authors

The manuscript titled "QSMs: interactions between plant and the associated pathogens" is a review article explores quorum sensing (QS) mechanisms in plant-microbe interactions, focusing on quorum sensing molecules (QSMs) produced by bacteria, fungi, and parasitic nematodes. The review details how plants perceive and respond to these QSMs and employ strategies to disrupt pathogen communication, emphasizing ecological and agricultural implications. The manuscript is well-structured, timely, and offers valuable insights into molecular biology, plant pathology, and sustainable agriculture.

The review systematically addresses QSMs from bacteria (e.g., AHLs, AI-2, DSF, AIPs), fungi (e.g., farnesol), and nematodes (e.g., ascarosides), outlining their functions and plant responses. This broad scope is a key strength. 

The abstract is concise but lacks specificity: adding a sentence highlighting a key finding or novel contribution would strengthen its impact. 

Section 4: Perceive and Respond to QSMs: This section effectively details plant responses but uses technical terms (e.g., "MAPK cascades," "PTI") without definitions, which may confuse non-specialists. Brief explanations would broaden readability.

Section 5: Strategies of Plants to Disrupt Pathogen QSMs: The section outlines plant countermeasures well but lacks concrete examples. Adding one or two case studies (e.g., a plant species and its QSI compound) would improve clarity and engagement.

Section 6: Plant-mediated QS Regulation and Ecological Implications: This dense section ties the review together but would benefit from subsections (e.g., "Metabolic Responses," "Ecological Impacts," "Agricultural Applications") for better readability.

Author Response

Reviewer #2: The manuscript titled "QSMs: interactions between plant and the associated pathogens" is a review article explores quorum sensing (QS) mechanisms in plant-microbe interactions, focusing on quorum sensing molecules (QSMs) produced by bacteria, fungi, and parasitic nematodes. The review details how plants perceive and respond to these QSMs and employ strategies to disrupt pathogen communication, emphasizing ecological and agricultural implications. The manuscript is well-structured, timely, and offers valuable insights into molecular biology, plant pathology, and sustainable agriculture.

Response: Dear reviewer 2, thank you very much for your attention to our manuscript. The pertinent suggestions you provided are extremely helpful in enhancing the quality of our manuscript. We have made a thoroughgoing revision of the manuscript based on your suggestion. Please refer to the revised manuscript for more details.

Comment 1. The abstract is concise but lacks specificity: adding a sentence highlighting a key finding or novel contribution would strengthen its impact.

Response: In response to your constructive feedback, we have highlighted the key creative in the abstract. As follows “Most significantly, the attention is given to the role of fungal and nematode QS molecules in modulating plant microbe interactions”.

Comment 2. Section 4: Perceive and Respond to QSMs: This section effectively details plant responses but uses technical terms (e.g., "MAPK cascades," "PTI") without definitions, which may confuse non-specialists. Brief explanations would broaden readability.

Response: we have added the Brief explanations for the technical terms (e.g., "MAPK cascades," "PTI"). Besides, we inserted Table 2 to make our glossary abbreviation easily understood by the reader.

Table 2 abbreviation

Term

Definition

DSF

Diffusible signaling factor

QS

Quorum sensing

QSM

Quorum sensing molecular

QSI

Quorum sensing inhibitor

PTI

Pattern-triggered immunity

MAPK Cascades

Mitogen-activated protein kinases Signaling pathways

WRKY/MYB

Plant transcription factor families

AHL

Acyl-homoserine lactone

MAMP

Microbe-associated molecular pattern

PAMP

Pathogen-associated molecular patterns

JA

Jasmonic acid

SA

Salicylic acid

Comment 3. Section 5: Strategies of Plants to Disrupt Pathogen QSMs: The section outlines plant countermeasures well but lacks concrete examples. Adding one or two case studies (e.g., a plant species and its QSI compound) would improve clarity and engagement.

Response: We thank the reviewer for highlighting this point. To improve the section’s engagement and clarity, we have added two detailed cases in Section 5 (QS Mimics).

Added in Section 5.1 (Lines 304-309):

“The chemical structure of AHLs is inherently unstable, making them susceptible to hydrolysis or acylation in the environment or by plant-released hydrolytic enzymes such as acylases or lactonases[111-113]. Moreover, plant-derived exudates: alkaloids, sugars, hormones, polysaccharides, proteins, and lactones, exhibit notable anti-plant pathogenic bacteria activity[114]. A. thaliana growth effects depend on AHL amidolysis by a plant-derived fatty acid amide hydrolase (FAAH), yielding l-homoserine, which can encourage plant growth at low concentrations by stimulating transpiration[111]. Moreover, plants can edit parasitic nematode-derived Ascr#18 pheromones to produce other ascaroside groups involved in developing defense mechanisms[108].”

And Section 5.2 (line 311-314)

“QSIs that target AI signaling molecules are primarily AHL-lactonases (for example, autoinducer inactivation A, AiiA hydrolyzes 3-oxo-C14-HSL in rice blight resistance), oxidoreductases, neutralizing antibodies (such as mAb AP4-24H11 binding to S. aureus AIP-1) and small molecules such as vanillin (which blocks tomato C8-HSL receptor binding) [115-117]”

Comment 4. Section 6: Plant-mediated QS Regulation and Ecological Implications: This dense section ties the review together but would benefit from subsections (e.g., "Metabolic Responses," "Ecological Impacts," "Agricultural Applications") for better readability.

Response: we have analyzed the content suggested by the reviewers in lines 391-422. And we divided the section 6 into three subsections according to your suggestions. Please review the resubmitted manuscript.

Once again, thank you very much for your arduous work and instructive suggestions.

Yours sincerely

Xin Wang

Reviewer 3 Report

Comments and Suggestions for Authors

The review paper discusses various signalling molecules of bacteria, fungi and nematodes, thereafter focuses on the interactions or responses of these on plants. Based on this study, it provides a collation of information based on literature and no indepth analysis of this info was done to include various perspectives as authors understanding. More indepth knowledge lies on the "how" the mentioned signalling molecules regulate (at molecular and structural level) the suggested proccesses. The paper would be more appreciated should it delve more here, otherwise as it is, it's more on the surface level. 

Since it is a review paper, it appears to lack creativity. 

Title is appears vague and authors could consider being creative after careful reading and considering the context of the review and message conveyed

English editing to improve the paper is recommedended. 

The other downfall include repetition, particularly the introductory section and section 2 and 3. It is rather advisable to review the sections to omit repetition, and possibly consider collation of section 2 and 3, while removing redundancy. 

Section 4 lacks the "how" and not well discussed. 

How do plants recognise the short chain vs medium vs long chain molecules for the triggering of these beneficial reponses. 

Authors wrote with a lot of assumption, noted based the use of abbreviations without first full mention of the terminologies. 

Comments on the Quality of English Language

Highly recommended. 

Author Response

Reviewer #3: The review paper discusses various signalling molecules of bacteria, fungi and nematodes, thereafter focuses on the interactions or responses of these on plants. Based on this study, it provides a collation of information based on literature and no indepth analysis of this info was done to include various perspectives as authors understanding. More indepth knowledge lies on the "how" the mentioned signalling molecules regulate (at molecular and structural level) the suggested proccesses. The paper would be more appreciated should it delve more here, otherwise as it is, it's more on the surface level.

Response: Dear reviewer 3, we sincerely appreciate your insightful evaluation of our manuscript and your constructive critique regarding its current limitations. We have made a thoroughgoing revision of the manuscript based on your suggestion. Please refer to the revised manuscript for more details.

Comment 1. Title is appears vague and authors could consider being creative after careful reading and considering the context of the review and message conveyed.

Response: As your suggestion, we have attempted to change the title to "Quorum Signaling Molecules: Interactions Between Plants and Associated Pathogens."

Comment 2. English editing to improve the paper is recommedended.

Response: We have carefully double-checked our manuscript and modified the question you mentioned. Meanwhile, we invited professional editors to make progress in the language. MDPI Author Services (ID: english-94088) .

Comment 3. The other downfall include repetition, particularly the introductory section and section 2 and 3. It is rather advisable to review the sections to omit repetition, and possibly consider collation of section 2 and 3, while removing redundancy.

Response: Thank you for your valuable feedback. We have removed the definitions of QS and the classifications of QSMs in the introduction section (e.g. lines 101-109 in the previous manuscript). We also deleted some repetitions in other sections.

Comment 4. Section 4 lacks the "how" and not well discussed.

Response: Thank you for your feedback regarding Section 4, "Perception of and Responses to QSMs." We recognize that the original version of this section lacked sufficient detail on the mechanisms underlying plant perception and responses to quorum sensing molecules (QSMs). In the revised manuscript, we have significantly expanded this section to provide a more comprehensive explanation of the "how" behind these processes. Specifically, we have included detailed descriptions of the molecular mechanisms and signaling pathways involved in plant recognition of QSMs. For bacterial N-acyl homoserine lactones (AHLs), we now elaborate on how plants detect these molecules based on their chain lengths: Short-chain AHLs (e.g., C4-HSL, C6-HSL): These small, hydrophobic molecules diffuse passively across plant cell membranes and interact with cytosolic receptors, such as histidine kinases (e.g., HK1), initiating growth-promoting responses via auxin and cytokinin signaling. Medium-chain AHLs (e.g., C8-HSL, C10-HSL): These are recognized by membrane-bound receptors, including G-protein-coupled receptors (GPCRs) like GCR1-GPA1 and receptor-like kinases (RLKs) such as Cand2/Cand7, triggering MAPK cascades and calcium signaling to balance growth and immunity. Long-chain AHLs (e.g., C12-HSL, C14-HSL): These require active transport into cells via ABC transporters (e.g., ABCG40) and are detected by lectin receptor kinases (e.g., LecRK-I.9), leading to immune priming through MAPK activation, ROS bursts, and epigenetic modifications.

Additionally, we have enhanced the discussion on fungal and nematode QSMs: Fungal QSMs (e.g., farnesol): We detail their recognition by oxysterol-binding protein (ORP) homologs and GPCR-like receptors with steroid-binding domains, which modulate SA or JA/ET defense pathways. Nematode QSMs (e.g., ascr#18): We describe their detection by the NILR1 receptor, followed by peroxisomal β-oxidation, which generates immune-activating derivatives. These revisions, supported by updated subsections (e.g., 4.1 "Molecular Mechanisms of QSM Perception in Plants"), aim to clarify the "how" by outlining specific receptors, transport mechanisms, and downstream signaling cascades. We believe this addresses the concern and provides a more robust discussion of plant-QSM interactions.

Comment 5. How do plants recognise the short chain vs medium vs long chain molecules for the triggering of these beneficial responses. 

Response: Thank you for raising this critical question regarding how plants discern short-, medium-, and long-chain molecules to activate distinct beneficial responses. To address this, we have significantly expanded Section 4.2 to delineate the chain-length-dependent mechanisms of AHLs and other signaling molecules. Specifically, short-chain AHLs (e.g., C4-HSL, C6-HSL) enhance growth by modulating auxin/cytokinin signaling and cell cycle regulators (e.g., YUCCA, PIN, CYCD), albeit with mild immune suppression, while long-chain AHLs (e.g., C12-HSL, C14-HSL) prime immunity via epigenetic reprogramming (H3K4me3, H3K9ac) and preactivation of defense pathways (MAPK cascades, ROS/NO bursts), prioritizing SA/JA signaling over growth-related processes. Medium-chain AHLs (e.g., C8-HSL, C10-HSL) exhibit dual roles: at low concentrations, they promote growth through GPCR-mediated coordination of auxin, cytokinin, and strigolactone pathways (e.g., AtMYB44), whereas higher concentrations trigger immune gene expression (e.g., PR1, WRKY53), reflecting their ecological versatility.

Beyond AHLs, we clarify that non-diffusible signals like AIPs rely on ABC transporters and two-component phosphorelay systems for extracellular perception, contrasting with diffusible DSF molecules (e.g., XcDSF). The latter exhibit concentration-dependent duality: at low levels (25 μM), XcDSF subverts PTI by hijacking sterol biosynthesis and disrupting FLS2 clustering, while higher concentrations (100–1000 μM) elicit robust immunity (callose deposition, HR-like cell death). These revisions, supported by new mechanistic schematics (Figures 5-6), underscore how structural features (chain length, diffusibility) and spatiotemporal dynamics govern plant-microbe crosstalk. We sincerely appreciate your guidance and welcome further suggestions to refine this framework.

Comment 6. Authors wrote with a lot of assumption, noted based the use of abbreviations without first full mention of the terminologies.

Response: We sincerely appreciate your invaluable feedback regarding the potential ambiguity in terminology, which we recognize could indeed lead to reader confusion. In response, we have thoroughly revised the manuscript to address these concerns: ambiguous terms and assumptions have been rephrased with additional citations for clarity (highlighted in red in the resubmitted version), and some abbreviations—including QS ("quorum sensing"), QSM ("quorum sensing molecules"), and AHL ("N-acyl homoserine lactone")—are explicitly defined upon their first mention in the Abstract, Section 1, and subsequent sections. Critical terms such as GPCR (G-protein-coupled receptor), RLK (receptor-like kinase), MAPK (mitogen-activated protein kinase), and ROS (reactive oxygen species) now follow this standardized practice. To further mitigate misinterpretation, we have added Table 2 at the end of the manuscript, serving as a glossary for key abbreviations (e.g., DSF: Diffusible signaling factor; PTI: Pattern-triggered immunity; JA: Jasmonic acid), thereby eliminating reliance on prior terminology familiarity. Should any residual ambiguities remain, we would be deeply grateful for your further guidance to refine the manuscript to meet the highest scholarly standards.

Once again, thank you very much for your arduous work and instructive suggestions.

Yours sincerely

Xin Wang

Round 2

Reviewer 2 Report

Comments and Suggestions for Authors

The revised manuscript, titled "Quorum Signaling Molecules: Interactions Between Plants and Associated Pathogens," has undergone significant improvements and is now ready for publication. The authors have thoroughly addressed the reviewers' comments, resulting in a well-organized, comprehensive, and scientifically robust review article.

The manuscript provides a detailed and systematic examination of quorum sensing (QS) mechanisms across bacteria, fungi, and nematodes, focusing on their interactions with plants. The revisions have clarified the roles of quorum sensing molecules (QSMs) in plant-pathogen dynamics, with enhanced sections on molecular mechanisms, plant responses, and agricultural applications. Overall, the revised manuscript is scientifically sound, well-written, and makes a significant contribution to the field of plant-microbe interactions. It is ready for publication and will serve as a valuable resource for researchers in molecular biology, plant pathology, and sustainable agriculture.